# EVALUATING GRAPHICAL PERCEPTION OF LARGE MULTIMODAL MODELS

## ABSTRACT

Despite the promising results of large multimodal models (LMMs) in various vision-language tasks, recent benchmarks reveal that these models can struggle with low-level chart perception tasks that require precision. However, since existing benchmarks primarily focus on end tasks that evaluate models' knowledge and reasoning abilities all together, they provide limited fine-grained insights into how the models' perception abilities affect their performance in chart tasks. To address this gap, we leverage *the theory of graphical perception*, an approach used to study how humans decode visual information encoded on charts and graphs, to develop an evaluation framework for analyzing gaps in LLMs' perception abilities in charts. With automated task generation and response evaluation designs, our framework enables comprehensive and controlled testing of LMMs' graphical perception across diverse chart types, visual elements, and task types. We apply our framework to evaluate the perception capabilities of state-of-the-art LMMs at three granularity levels (chart, visual element, and pixel). Our findings underscore several critical limitations of current state-of-the-art LMMs, including GPT-4o: their inability to (1) generalize across chart types, (2) understand fundamental visual elements, and (3) cross reference values within a chart. These insights provide guidance for future improvements in perception abilities of LMMs. The evaluation framework and labeled data will be publicly available upon acceptance.

## 1 INTRODUCTION

Large multimodal models (LMMs; OpenAI (2024); Gemini Team (2023)) have shown impressive results in a range of visual-language tasks (Yue et al., 2024a; Lu et al., 2024), including complex knowledge- and reasoning-intensive tasks over infographics in scientific documents. However, recent studies (Wang et al., 2024), along with our findings, have revealed limitations in their ability to perform low-level perception tasks, such as retrieving specific values or finding the extremum in given charts, as evidenced by Figure 1 which shows an example used in our framework.

Such an inconsistency indicates that existing LMMs may have potential limitations in their perception abilities. Yet, there is a lack of systematic exploration of these limitations. Existing benchmarks primarily focus on overall task performance, often combining perception, knowledge, and high-level reasoning into a single accuracy score. This metric serves as an indirect proxy for measuring LMMs' perception of charts—a high score may suggest good perception abilities, but poor performance makes it unclear whether the failure stems from perception errors, lack of knowledge, reasoning flaws, or a combination of these factors. Additionally, models may not even need to perceive and reason about charts to answer questions: as reported in recent studies, models can generate correct answers without the visual input (Yue et al., 2024b; Chen et al., 2024a). Finally, the way current models perceive charts—especially their understanding of fundamental visual elements—remains unexplored in existing evaluations. Thus, it is desirable to systematically study models' perception capabilities to understand factors limiting their performance in low-level chart perception tasks.

In this paper, we leverage *graphical perception* (Cleveland & McGill, 1984), a theory originally developed to study human interpretation of visual data in charts and graphs, to evaluate models' perception capabilities. For example, prior studies on human graphical perception show that, because humans can better perceive the length of lines than area sizes when comparing values, we can more efficiently and accurately read bar charts than pie charts to answer questions about calculating the

differences between two values based on their visual representation in the charts. This motivates us to evaluate models' fundamental graphical perception (e.g., perception of color, length, size) to explore the limitations of current LMMs. To achieve this goal, we aim to test LLMs' performance on a range of chart perception tasks that involve reading and interpreting data based on its visual representation across a diverse set of chart types (e.g., bar, line, scatter, pie) and visual elements (e.g., color, length, size). This approach could offer a direct and comprehensive evaluation of models' perceptual abilities, especially helping us understand in what aspects the models fail to generalize.

We introduce an evaluation framework specifically designed to assess the graphical perception abilities of state-of-the-art (SOTA) LMMs (OpenAI, 2024; Chen et al., 2024b; Abdin et al., 2024; Meng et al., 2024). Our framework includes an automated task generation and response evaluation pipeline that synthesizes a diverse set of chart perception tasks with different chart representations from a set of seed datasets, allowing us to scale up the evaluation with minimal human intervention. With this framework, we explore the perception abilities of SOTA LMMs in a coarse-to-fine manner, ranging from chart-type-level performance to the fundamental visual elements forming the charts, and to the pixel-level analysis that reveals how models perceive specific regions in the charts. Our goal is to understand where and how models fail to generalize in their perception of charts. Our research questions and key findings are listed as follows. These findings provide fine-grained insights into the low-level visual abilities of LMMs from the perspective of graphical perception.

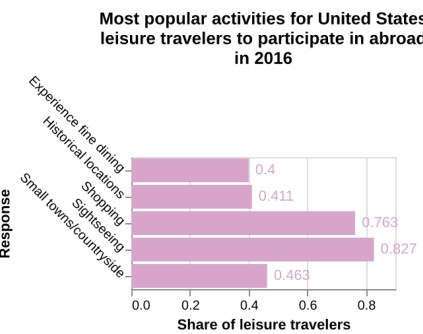

Figure 1: Given *"Identify the activity with the highest share of leisure travelers."* on this chart, GPT-4o answers *"Small towns/countryside"* in 10 out of 10 responses. The correct answer is *"Sightseeing"*.

**RQ1: Can SOTA LMMs Generalize Across Diverse Chart Types?** Our analysis reveals that LMMs exhibit significant performance fluctuations depending on the chart type and rely heavily on the explicit numerical annotations. This demonstrates their lack of generalization across different chart types, despite the simplicity and identical information presented in these charts.

**RQ2: Do LMMs Learn Generalizable Graphical Perception Beyond Chart Patterns?** We find that LMMs perform relatively well only on charts with specific combinations of visual elements (e.g., length, size, position) but struggle to generalize to charts composed of similar visual elements. This indicates that current LMMs fail to understand the fundamental and generalizable visual elements within various charts, instead learning to perceive only superficial chart patterns.

**RQ3: Where Do LMMs Fall Short in Pixel-Level Understanding of Charts?** We find that while models often successfully locate important regions required for solving simple tasks such as retrieving data point values, their referencing of these values is frequently imprecise, leading to only approximate outputs. This imprecision accumulates in more complex tasks, such as ordering all the data points, resulting in significant errors due to error propagation.

## 2 EXPERIMENTAL SETUP

The major difference between prior chart benchmarks (Masry et al., 2022; Wang et al., 2024) and ours is that we do not aim to create a more challenging benchmark. Instead, as shown in Figure 2, we focus on *automatically* creating the *simplest* possible tasks to evaluate and diagnose the graphical perception abilities of current SOTA LMMs using diverse charts and visual elements.

### 2.1 DATASETS

For our evaluation, we utilize the VisText (Tang et al., 2023) dataset as the primary data source as it covers diverse data domains including sports, news, finance, health, etc. More importantly, it includes both textual data tables and Vega-Lite programs (Satyanarayan et al., 2017) that generate rasterized charts after our light-weight editing, making it an ideal starting point for customized chart and task generation, as well as the following automated evaluation. We randomly sample a total of 1,000 datasets from VisText, ensuring a wide variety of data types and relationships are represented.

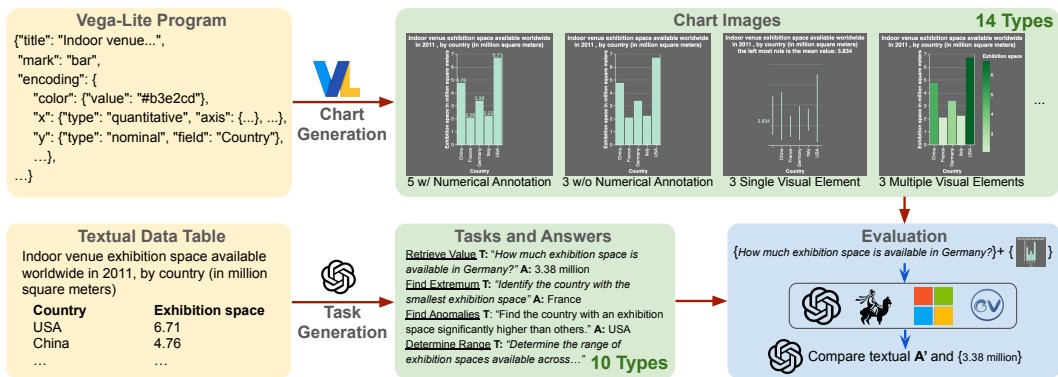

Figure 2: Framework of data synthesis and evaluation. With randomly sampled 1,000 datasets as seeds, we edit the Vega-Lite program to generate 14 types of charts and use GPT-4o with textual data tables to generate 10 types of tasks and corresponding answers, resulting in a total of 140,000 inputs for each model to be evaluated. For evaluation, we consider the most representative models from four model categories and their responses are automatically evaluated by GPT-4o in text format.

The datasets we use include three major types of data attributes: (1) Nominal Attributes: Categorical variables that represent distinct labels without an inherent order (e.g., country, movie genres). (2) Ordinal Attributes: Variables that have a meaningful order or ranking but no fixed intervals between values (e.g., movie ratings, years). (3) Numerical Attributes: Continuous variables that allow for the calculation of differences and other mathematical operations (e.g., exhibition space in Figure 2). To ensure simplicity, each dataset has at most two data dimensions (i.e., a nominal attribute paired with a numerical attribute or an ordinal attribute paired with a numerical attribute). In addition, the number of data points is limited to 5. These constraints allow us to evaluate the models' graphical perception capabilities without overwhelming them with complex, multidimensional data.

## 2.2 TASKS

To automatically generate tasks from seed datasets, we prompt GPT-4o with (1) textual data tables, (2) chart descriptions (Vega-Lite programs), and (3) task types, asking it to instantiate tasks based on the data and descriptions, and ensuring that the tasks are answerable using the chart image rendered from the Vega-Lite program. During model inference, only chart images and task texts are provided to the model. Following previous work (Saket et al., 2019) for evaluating human perception, we design 10 types of common tasks for each dataset, as shown in Table 1.

Table 1: All 10 task types, ranging from a single data point (T1) to an entire dataset (T10).

| Task | Description |
|------|-------------|
| T1. Retrieve Value | Retrieve the value of a given attribute for a specific data point. |
| T2. Find Extremum | Identify the maximum or minimum value of a specified attribute. |
| T3. Find Anomalies | Detect anomalies in the dataset regarding a given relationship or expectation. |
| T4. Determine Range | Determine the range of values for a given attribute. |
| T5. Find Correlation | Identify any correlation between two data attributes. |
| T6. Compute Derived Value | Compute a derived value from a set of data points. |
| T7. Filter | Filter the data points based on specific conditions. |
| T8. Order | Order the data points according to a numerical attribute. |
| T9. Find Clusters | Find clusters of similar attribute values. |
| T10. Characterize Distribution | Characterize the distribution of a data attribute over a given set. |

These tasks are designed to cover a broad spectrum of graphical perception skills, ranging from a single data point (e.g., T1), to multiple data points (e.g., T4), and to an entire dataset (e.g., T10). This design allows us to evaluate how well models handle increasing levels of task complexity. Please refer to Appendix A for the detailed task generation prompt.

## 2.3 MODEL SELECTION

Given the extensive variety of LMMs with different vision and language backbones, a comprehensive evaluation and fair comparison of all models may not be feasible. Therefore, we focus on four categories of models: proprietary, open-source, lightweight, and chart-specialized LMMs. We select the most representative model from each category for evaluation, based on the averaged results reported on prior chart-included benchmarks (Wang et al., 2024; Yue et al., 2024a; Lu et al., 2024).

- **GPT-4o** (OpenAI, 2024), the strongest proprietary general-purpose model, represents the SOTA in LMMs. Benchmarking GPT-4o allows us to evaluate the performance of the latest model in chart tasks, providing a reference point for comparison with other models in this domain.
- **InternVL2** (Chen et al., 2024b) is one of the best open-source general-purpose LMMs. It is built upon Llama3.1 (Dubey et al., 2024) and has a total of 76B parameters. Evaluating InternVL2 can show the gap between open-source models and GPT-4o in graphical perception.
- **Phi-3.5-Vision** (Abdin et al., 2024) is selected as a strong lightweight general-purpose LMM, with only 4.2B parameters. With Phi-3.5, we can evaluate whether models with smaller vision backbones can still maintain decent levels of graphical perception.
- **ChartAssistant** (Meng et al., 2024) is the best chart-specialist model. It is continually trained with the LLaVa-13B (Liu et al., 2023) on a massive amount of chart datasets, including the original VisText dataset. With this specialist model, we can measure the benefits of in-domain training in enhancing perception and generalization abilities.

## 2.4 EVALUATION

We employ GPT-4o as an automated text evaluator, which is particularly useful when models being evaluated output varied answer formats, such as Chain-of-Thought reasoning format (Wei et al., 2022), or when dealing with open-ended tasks. GPT-4o evaluates responses by comparing the textual responses of models against the predefined answer, which is generated automatically by GPT-4o based on textual representation of the data and chart program that do not need visual perception. The evaluation process is guided by a detailed rubric designed for different task types. For example, in *Retrieve Value* tasks, answers are considered accurate if they are within a 5% margin of the correct value. For order-based tasks, such as ranking items, the model must return the exact sequence expected, while other list-based tasks do not require specific ordering. Evaluation outcomes are categorized into accurate, fair, skipped, inaccurate, and n/a. See Appendix B for details.

To calibrate GPT-4o's evaluation process, we use a 10-shot demonstration (Brown et al., 2020) that includes examples of textual data tables, tasks, reference answers, model responses, and expected evaluations. This calibration helps ensure consistency and accuracy in evaluation. We validate the reliability of GPT-4o as an evaluator by manually checking 200 evaluation results of GPT-4o and InternVL2, achieving an accuracy rate of 99.5% without noticeable bias across different models. This establishes a reliable foundation for our evaluation framework.

## 3 RQ1: CAN SOTA LMMS GENERALIZE ACROSS DIVERSE CHART TYPES?

> **Takeaways: No**
>
> - Despite showing decent performance on specific chart types, LMMs struggle with variants of the same charts, showing limited generalization.
> - LMMs heavily rely on explicit numerical annotations, performing significantly worse when annotations are removed.

In this section, we analyze the performance of models at the chart-type level, where we compare models' performance in solving tasks with data represented in different chart types (line, bar, scatter, with and without explicit numerical annotations). Despite the data being presented differently across charts, the models are expected to achieve similar performances due to the simplicity of the charts. Figure C1 shows chart examples used in this section.

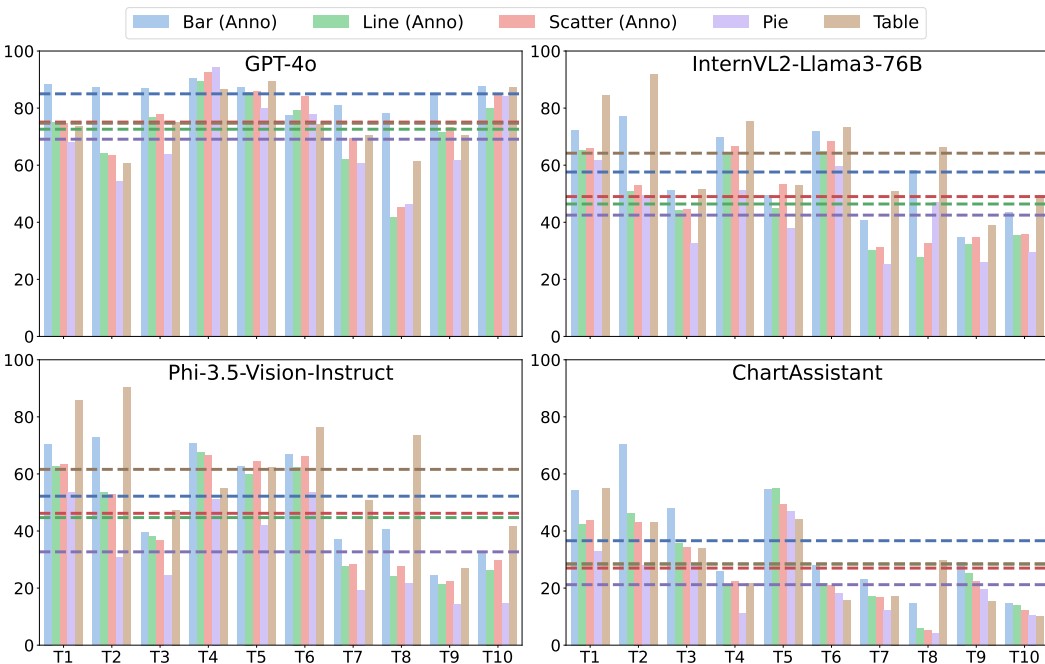

Figure 3: Accuracy of models on different types of charts with numerical annotations given the same 10 types of tasks. The dotted line refers to the average performance by chart type and color refers to the given chart type, and T-$i$ indicates the $i$-th task detailed in Section 2.2.

## 3.1 ANALYSIS ON CHARTS WITH NUMERICAL ANNOTATIONS

Figure 3 presents the performance of models on different types of charts with numerical annotations given the same tasks, from which we can make the following observations:

(1) A significant performance gap exists for the same model when interpreting different types of charts containing the same information, suggesting that these models lack generalization across chart types. For example, GPT-4o consistently performs the best overall, yet shows a clear preference for Bar (Anno) over Pie charts (85.0% vs. 69.1%). This indicates that while GPT-4o excels in understanding some chart types, it still relies on specific visual structures to achieve its highest performance. Meanwhile, InternVL2 and Phi-3.5, the open-source general-purpose models, perform best when presented with table images. This observation suggests that these models might be specifically optimized for structured data. However, the performance gaps of these models between different chart types are even larger, with up to 21.7% for InternVL2 and 28.9% for Phi-3.5, highlighting a stronger dependency on specific visual structures.

(2) Despite being trained on an extensive range of chart-related datasets and tested on simplified versions of its training data, ChartAssistant underperforms general-purpose models and struggles to generalize effectively across different chart types. Its relatively better performance on Bar (Anno) can be attributed to the fact that bar charts make up 44.3% of its training data. This raises concerns about the effectiveness of chart-specific training for generalization.

(3) Models demonstrate significant performance variations across the ten tasks (T1 to T10), and task complexity amplifies the inconsistencies across chart types. For example, GPT-4o performs relatively consistently across all tasks when given Bar (Anno) charts. However, when interpreting Pie charts, its performance varies dramatically, with a gap of up to 48% between simpler tasks like *Determine Range* (T4, 93%) and more complex tasks like *Order* (T8, 46.3%). A similar trend can also be observed in the other two general-purpose open-source models.

These observations highlight the importance of improving graphical perception across a broader range of chart types to enhance LMMs' generalization in real-world applications where diverse charts appear in various forms. Please refer to Appendix D for more detailed results.

## 3.2 ANALYSIS ON CHARTS WITHOUT NUMERICAL ANNOTATIONS

Table 2: Overall accuracy of models given the charts with and without explicit numerical annotations. Exemplar charts are shown in Figure 2 (Chart Images) and Figure C1.

| | **Bar** | | | **Line** | | | **Scatter** | | |
| | w/ Anno. | w/o Anno. | Δ | w/ Anno. | w/o Anno. | Δ | w/ Anno. | w/o Anno. | Δ |
|---|---|---|---|---|---|---|---|---|---|
| GPT-4o | 85.0 | 53.4 | -31.6 | 72.6 | 42.8 | -29.8 | 75.1 | 41.2 | -33.9 |
| InternVL2 | 57.6 | 45.9 | -11.7 | 46.4 | 33.0 | -13.4 | 49.0 | 33.4 | -15.6 |
| Phi-3.5 | 52.2 | 32.4 | -19.8 | 44.7 | 26.0 | -18.7 | 46.2 | 27.9 | -18.3 |
| ChartAssistant | 36.6 | 33.9 | -2.7 | 28.5 | 25.9 | -2.6 | 27.0 | 25.4 | -1.6 |

Table 2 shows the performance differences of models when transitioning from charts with numerical annotations (w/ Anno.) to charts without annotations (w/o Anno.). GPT-4o shows the most significant drop in performance across all chart types when numerical annotations are removed, with an average performance decrease of 31.8% across the three chart types. This indicates that GPT-4o, despite being a leading model, still struggles to accurately perceive charts without the aid of numerical annotations. Similarly, Phi-3.5 and InternVL2 exhibit substantial performance declines.

Additionally, Phi-3.5 shows a greater decline in performance compared to InternVL2 (e.g., -18.9% vs. -13.6% on average), demonstrating that lightweight LMMs have weaker generalization abilities than larger models when faced with charts lacking explicit numerical cues. These observations show the importance of developing LMMs that are less reliant on numerical annotations as many complex charts in real-world scenarios do not include such annotations. See Appendix D for detailed results.

## 4 RQ2: DO LMMS LEARN GENERALIZABLE GRAPHIC PERCEPTION BEYOND CHART PATTERNS?

> **Takeaways: No. Superficial Chart Patterns Only**
>
> - LMMs achieve relatively decent performance only when given specific combinations of visual elements but struggle even when generalizing to very similar charts, showing their lack of robust understanding of fundamental and generalizable visual elements.

Visual elements (Bertin, 1967; Cleveland & McGill, 1984; Munzner, 2014) are the core building blocks of data visualization, defining how quantitative values in charts are visualized. Following prior work on human, we use four fundamental visual elements that are widely used to represent data values in charts: the position of a point (e.g., the top part of a bar), the length of a rule (e.g., bars or lines), the size of a region (e.g., the area of a bar), and the saturation of a color. By systematically analyzing models' results on charts composed of these elements, we aim to assess how each element—or a combination thereof—impacts model perception, identifying which visual elements are most effective or challenging for current models. Particularly, as some of the generated charts may not be common, we provide detailed guidelines on how to interpret these charts for LMMs.

Table 3: Overall accuracy of models given charts rendering values with single or multiple visual elements. For multiple-element charts, the same value is redundantly encoded through different elements. For example, the size, top part, and length of a bar are all proportional to the values.

| | Single Element | | | | Multiple Elements | | | |
| | Length (↔) | Color (▩) | Size (◉) | Position (⋆) | ↔, ⋆ | ◉, ⋆ | ↔, ◉, ⋆ | ↔, ▩, ◉, ⋆ |
| Toy Chart |  |  |  |  |  |  |  |  |
| GPT-4o | 17.6 | 21.1 | 22.6 | **41.2** | 22.9 | 24.4 | **53.4** | 27.7 |
| InternVL2 | 18.3 | 20.7 | 21.1 | **33.4** | 25.2 | 24.1 | **45.9** | 26.5 |
| Phi-3.5 | 17.8 | 18.9 | 19.6 | **27.9** | 21.2 | 20.9 | **32.4** | 22.7 |
| ChartAssistant | 12.3 | 14.0 | 13.6 | **25.4** | 19.2 | 17.4 | **33.9** | 18.5 |

Table 3 presents the performance of models when interpreting charts rendered with single or multiple basic visual elements. We make three observations:

(1) LMMs suffer from basic visual element understanding. Across the board, models show relatively poor performance when interpreting charts that rely on a single visual element, such as length, color, or size. For example, GPT-4o achieves only 17.6% accuracy on charts using length alone, despite its otherwise strong performance. This indicates a fundamental challenge for LMMs in extracting quantitative values from basic visual elements, potentially limiting their abilities when comprehending complex charts where such basic visual elements are used.

(2) Surprisingly, the addition of redundant visual elements often hurt model performance. For example, while using position only results in decent performance (e.g., GPT-4o scores with 41.2% accuracy), rendering values via size at the same time (●, ⋆) hurts the performance dramatically across all models. Although the size can be more straightforward than position for tasks like ordering, LMMs clearly fail to leverage the advantages of various visual elements in most of the times. This suggests that the presence of multiple visual elements may overwhelm the models' capacity to prioritize relevant visual cues, leading to confusion and misinterpretation of the data.

(3) LMMs often fail to generalize effectively across charts that use similar visual elements. For instance, models show strong performance on bar charts that combine position, length, and size (↔, ●, ⋆), but struggle with similar charts that only use position and length (↔, ⋆). This suggests that models excel only with specific combinations of visual elements and lack the robustness needed to transfer this understanding to simpler or slightly altered visualizations.

Overall, these results demonstrate that current LMMs merely follow specific and superficial perception patterns for common charts such as scatter (⋆) and bar (↔, ●, ⋆), while struggling to generalize beyond these familiar chart patterns. This highlights the necessity of improving models' understanding of fundamental visual elements beyond specific chart types, leading to better generalization and perception. Please refer to Appendix D for detailed results.

## 5   RQ3: WHERE DO LMMs FALL SHORT IN PIXEL-LEVEL PERCEPTION?

> **Takeaways: Imprecise Value Referencing**
>
> - LMMs often correctly localize the important regions in the bar charts for value retrieval (e.g., data points, axes), but they frequently fail to accurately cross reference the specific values, especially in charts without explicit number annotations.

To understand the perception mechanisms of LMMs, we conduct a pixel-level analysis to examine which specific regions of the charts that models attend to when generating responses. This analysis aims to test whether LMMs correctly attend to important regions and cross reference the values of the chart for the most basic *Retrieve Value* task.

### 5.1   METHODOLOGY

Faithfully interpreting transformer-based models remains an open problem (Bereska & Gavves, 2024; Singh et al., 2024), particularly in the context of newly emerging LMM capabilities. In our analysis, we seek to use techniques that are model-agnostic, i.e., they can be applied to a black-box model such as GPT-4o without access to its weights or activations.[1] The most popular model-agnostic interpretation techniques generally require many calls to the model with different corruptions (Ribeiro et al., 2016; Lundberg, 2017), which are computationally expensive for large LMMs like GPT-4o. We use a simplified, more efficient version of these methods that occludes different regions of the input image one at a time and measures the model's response. Specifically, we manually select 100 pairs of Bar and Bar (Anno) charts and label the important regions for the *Retrieve Value* task. Each image is divided into 144 non-overlapping regions and we corrupt each region one at a time by changing its pixels to the background color. We then calculate the difference in

---

[1]We also qualitatively explored some gradient-based and attention-based interpretation methods (Selvaraju et al., 2017; Wiegreffe & Pinter, 2019), but found that they were extremely sensitive to hyperparameters in the interpretation methods so we omit this analysis.

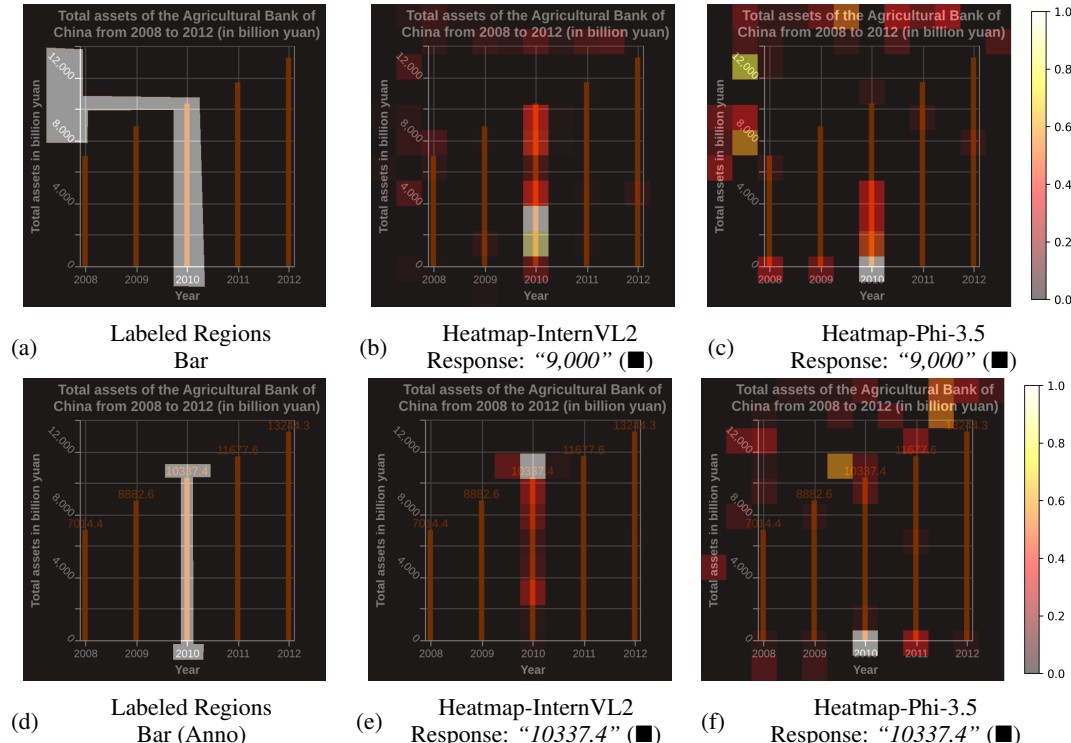

Figure 4: Examples of labeled regions and importance heatmaps for two models on Bar and Bar (Anno) charts. Given *"What is the value of total assets in billion yuan for the year 2010?"*, both models successfully locate most labeled important regions on both the Bar (Anno) and Bar charts but fail to reference the correct y-axis values on the Bar chart. The correct answer is *"10337.4"*.

generated token logits between the intact chart and the corrupted version for each model. We use the normalized logit difference as a measure of the feature importance of the region for the generated tokens. We aggregate these region-level feature importances into a heatmap and measure whether the high-importance regions cover most of the groundtruth labeled regions to determine whether models use these important regions to generate the response.

Table 4: Correctness at retrieving values in the *Retrieve Value* task (table rows) depends on whether an LMM correctly identifies important chart regions (table columns). Identifying important regions is measured by whether the groundtruth labeled regions are covered (■) or not covered (□) by the LMM's feature importance map for value retrieval. Important regions are successfully identified more often for Bar (Anno) charts (b & d). Sometimes, important regions are successfully identified but the model fails to retrieve the correct value (red).

|  | InternVL2 | | | | | | | Phi-3.5 | | | | | | |
|---|---|---|---|---|---|---|---|---|---|---|---|---|---|---|
|  | ■ | □ | | | ■ | □ | | | ■ | □ | | | ■ | □ |
| Correct | 52 | 2 | | Correct | 79 | 2 | | Correct | 29 | 6 | | Correct | 72 | 1 |
| Incorrect | 34 | 12 | | Incorrect | 13 | 6 | | Incorrect | 51 | 14 | | Incorrect | 23 | 4 |
| (a) Bar | | | | (b) Bar (Anno) | | | | (c) Bar | | | | (d) Bar (Anno) | | |

## 5.2 RESULTS

Table 4 shows that both InternVL2 and Phi-3.5 are quite effective at localizing important regions when given Bar (Anno) charts. As long as the models can identify regions with the correct numbers, they generally generate correct responses, proving their reliance on explicit number annotations for accurate value retrieval shown in Section 3. When annotations are removed (Bar), both models often still correctly locate the important regions but struggle to precisely refer to the values from the value-axis. Figure 4 illustrates this behavior. In the Bar (Anno) chart (Figures 4d-4f), both models accurately identify important regions and generate correct responses. In contrast, on Bar

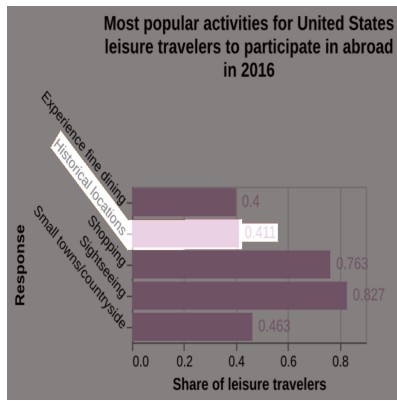

(a) Labeled Regions-Bar (Anno)

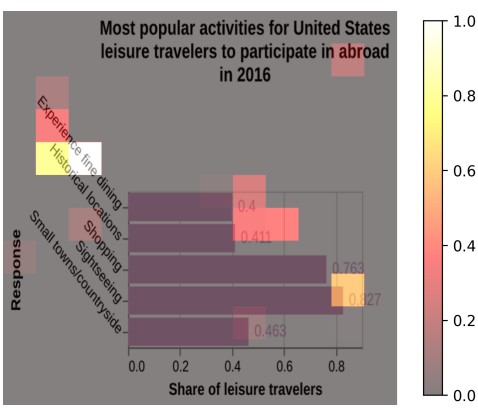

(b) InternVL2-Response: *"0.4"* (□)

Figure 5: Examples of labeled regions and importance heatmap of InternVL2 on a Bar (Anno) chart. Given the task *"Determine the share of leisure travelers for historical locations."*, InternVL2 incorrectly locates the bar for "Experience fine dining," which is closely positioned near the correct one. As a result, it generates an imperfect answer, 0.4. However, as this value is within 5% of the target value, 0.411, it is judged as correct according to the evaluation rubric.

charts (Figures 4a-4c), although they focus on the right areas, their responses are far from the correct values. In addition, Phi-3.5 tends to be more easily influenced by non-important regions compared to InternVL2, showing that lightweight models may be more sensitive to visual information irrelevant to the given task, leading to less favorable results shown in Figure 3.

Figure 5 demonstrates a case where models can produce correct answers even when not fully utilizing the important regions. This occurs when models focus on nearby data points that share similar values with the target data points, allowing them to approximate values closely enough to meet the evaluation criteria. This shows that, region localization abilities of LMMs tend to diminish when information is rendered unusually, such as when categories are shown obliquely.

## 6 DISCUSSION

In this section, we investigate the performance of SOTA LMMs when given more complex charts from the perspective of the number of data points and the data dimensions. Both results indicate that current LMMs are not capable of handling complex charts.

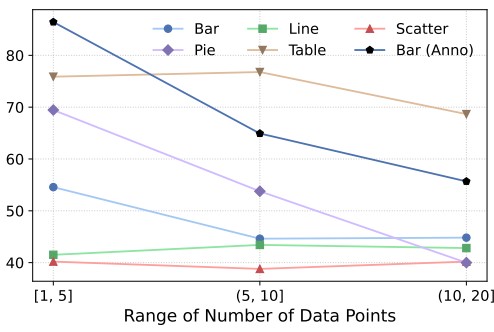

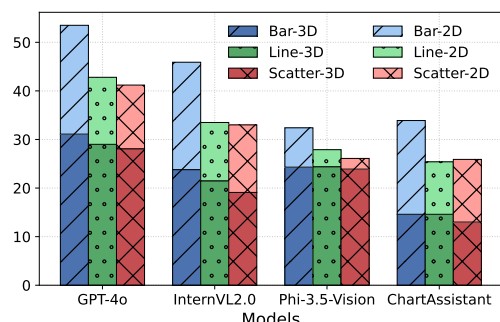

Figure 6: Overall accuracy of GPT-4o given 100 datasets with different sampled data points and different chart types.

Figure 7: Overall performance comparison of four LMMs when given two-dimensional and three-dimensional datasets.

### 6.1 PERFORMANCE CHANGES AS NUMBER OF DATA POINTS INCREASES

To measure the impact of the number of data points on LMMs' graphical perception abilities, we randomly sample 100 datasets, each containing at least 20 data points. We then systematically

reduce the number of data points into three buckets: $[1, 5]$, $(5, 10]$, and $[10, 20]$, and observe how model performance varies across common chart types representing these data points. Figure 6 shows the overall accuracy of GPT-4o when tested on 100 datasets. As the number of data points increases, the performance consistently declines across all chart types, highlighting the model's sensitivity to data density. Notably, Bar (Anno) charts exhibit the steepest drop in accuracy, suggesting that while numerical annotations aid graphical perception in simpler cases, the presence of more data points and numbers overwhelms the model's ability to effectively perceive the charts.

## 6.2 MULTI-DIMENSIONAL DATASET PERFORMANCE

We select 100 datasets from ChartLLM (Ko et al., 2024), ensuring each dataset contains three data dimensions (e.g., Nominal-Numerical-Nominal) with a controlled number of data points. These datasets are then manually edited to create the popular chart types of interest: bar, line, and scatter. Figure 7 compares the performance of various models when understanding two-dimensional (2D) and three-dimensional (3D) datasets across different chart types. The results indicate a notable performance drop when models are tasked with three-dimensional data visualization, particularly for bar and scatter charts. GPT-4o performs the best overall but still shows significant degradation when moving from 2D to 3D visualizations. InternVL2 and Phi-3.5 show similar trends, though Phi-3.5 is relatively more robust than other models. ChartAssistant performs poorly overall, with minimal adaptability between 2D and 3D contexts. These findings indicate that current LMMs cannot fully understand advanced data visualizations yet. Figure C3 shows examples of 3D charts.

## 7 RELATED WORK

**Graphical Perception**  Cleveland & McGill (1984) introduce graphical perception as the visual interpretation of data through basic visual elements, such as position, length, area, and color. Extensive research has since expanded on their work, evaluating human perception across diverse data types (Heer et al., 2009; Javed et al., 2010; Whitlock et al., 2020; Borkin et al., 2013), increasingly complex tasks (Saket et al., 2019; Xiong et al., 2023; Bearfield et al., 2024), and complex charts (Heer & Bostock, 2010). In the context of neural networks, prior work tests the graphical perception abilities of vision-only models (Haehn et al., 2019) following a similar protocol. However, evaluating and understanding the graphical perception of LMMs remains under-explored.

**Large Multimodal Models and Their Benchmarks**  Multimodal, especially vision-and-language, modeling has evolved significantly, beginning with early models (Tan & Bansal, 2019; Chen et al., 2020; Lu et al., 2019) that inject vision features into language understanding models, to those using contrastive learning for cross-modality representation (Radford et al., 2021; Yu et al., 2022; Zhang et al., 2024), and to the recent unified LMM frameworks (OpenAI, 2024; 2023; Gemini Team, 2023; 2024; Liu et al., 2024a; Cai et al., 2024) for various downstream tasks. Graphical perception is widely yet implicitly considered in benchmarks for evaluating these LMMs, including task-specific ones (Masry et al., 2022; Mathew et al., 2022; Wang et al., 2024) and the recent holistic ones (Yue et al., 2024a; Liu et al., 2024b; Lu et al., 2024; Yu et al., 2023). Despite its ubiquity, all existing benchmarks assess graphical perception indirectly, often evaluating it alongside other abilities like reasoning or by introducing increasingly complex charts. Our work, instead, uniquely isolates and directly evaluates the graphical perception abilities of LMMs in a comprehensive fashion.

## 8 CONCLUSION

This work introduces a comprehensive and configurable evaluation framework for automatically measuring the graphical perception abilities of LMMs, offering fine-grained insights into current state-of-the-art LMMs. Our findings reveal that these models struggle to generalize across diverse chart types, understand fundamental visual elements, and cross reference values within charts. In addition, our framework serves as a flexible test suite that can be easily adapted to support the development of future LMMs. Future work may leverage this framework to synthesize diverse data for training and testing on a wider range of tasks, potentially enabling improved graphical perception and general low-level visual reasoning. We hope these findings and the framework can guide the development of LMMs with more generalizable perception abilities in the future.

## ETHICAL AND REPRODUCIBILITY STATEMENT

This work focuses on evaluating LMMs' graphical perception abilities using a proposed automated framework applied to publicly available datasets. The findings in this paper aim to guide the development of LMMs with improved general perception, which can benefit various real-world applications. Therefore, we do not expect any major ethical issues arising from this research.

In terms of reproducibility, Section 2, Appendix A, and Appendix B provide detailed information on the experimental setup, task generation, and evaluation rubric. Upon acceptance, we will open-source our code, data, and evaluation framework to ensure full transparency and reproducibility.

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

# A  TASK GENERATION

---

**Task Generation Prompt (with text input only)**

You are a teacher to provide problems for students to solve. The problems are about understanding data and visualizations. We will provide you with an input data, a Vega-Lite program, and a task type that the understanding task should base on. You will need to design a chart understanding task contextualized in the given data and chart.

Design the task based off one of the following idioms:
• Retrieve Value. For this task, ask students to identify values of attributes for given data points. For example, what is the value of horsepower for Mazda CX50?
• Find Extremum: For given concrete conditions on data attribute values, ask students to find data points satisfying those conditions. For example, which car types have the most city miles per gallon?
• Find Anomalies: ask students to identify any anomalies within a given set of data points with respect to a given relationship or expectation. For example, which car types have abnormally low MPG?
• Determine Range: For a given set of data points and an attribute of interest, ask students to find the span of values within the set. For example, what is the range of car prices?
• Find Correlation: for a given set of two data attributes, ask students to determine if there is a correlation between them. For example, is there a strong correlation between car price and MPG?
• Compute Derived Value: for a given set of data points, ask students to compute an aggregate value of those data points. For example, what is the sum of the budget for the action and the sci-fi movies?
• Filter: For given concrete conditions on data attribute values, ask students to find data points satisfying those conditions. For example, which car types have miles per gallon ranging from 20 to 40?
• Order: For a given set of data points, ask students to rank them according to a specific ordinal metric. For example, list the car types based on their MPG from low to high.
• Find Clusters: for a given set of data points, ask students to count the number of groups of similar data attribute values. For example, how many different car brands are shown in the chart below?
• Characterize Distribution: for a given set of data points, ask students to identify the distribution of that attribute's values over the set. For example, what percentage of the cars with MPG higher than 30?

You need to match the following requirements:
  1. The task should be reasonable, and it should not exceed one sentence, and it should be contexualized in the given data.
  2. The task should be achievable by reading the visualization without referring other tools.
  3. The task should be self-contained with the given dataset, it should not require student to look up external information.
  4. Each task should have a standard answer, avoid generating questions like "compare two values of your choice."
  5. Try not to repeat the verb for each task to maximize diversity.

Create a `[Task]` based off the `[Data Summary]` and `[VegaLite Script]` provided.
The response should be in a json format:
`{"reason":...,"tasks":[{"description":...,"type":...},...]}`,    including how you design the task and the actual task description.
Generate 10 tasks at once.

**For example:**

```
[Data Summary]

|Date      |Location
0|5/12/2009|Houston, TX
1|4/18/2009|McAllen, TX
2|7/11/2009|Indianapolis, IN
3|11/14/2009|Kansas City, MO|MO
4|3/12/2010|Chicago, IL|IL
...

{Task Demonstration}
```

## B  EVALUATION RUBRICS

---

**Evaluation Prompt (with text input only)**

You are a teacher to grade students' answers. We will provide you a dataset, a list of tasks and student answers. Your goal is to use the dataset to evaluate if the student's answer is correct. In order to form a good judgement, you should first use the dataset to derive your answer, and then compare it with the students asnwer.

When you generate the referenece answer:
* If the task asks for a value, provide the value directly.
* If the task asks for trend or correlation, answer it with one of "increasing", "decreasing" if the general trend point to the direcrtion, otherwise provide "unclear".
* Provide a brief reasoning of how you come up with your answer in "reasoning" part.
* If you cannot answer a question, provide "I don't know" as the answer, try not to provide a wrong answer.

When evaluating student's answer:
The student_answer_correctness should include the grading results of the student's answer and must be one of the following options:
   - correct
   - fair (somewhat close but not precise)
   - incorrect
   - skipped (if the student skipped the answer)
   - n/a (if the task does not make sense or is not answerable with the given dataset)

Note that if the student's answer (value) is an approximation within 5% of your reference answer, it is considered as correct. If is is an approximation within 20% of your reference answer, it is fair.
For order-based tasks, such as ranking items, the student answer must match the expected orders you found. However, for list-based tasks where order is not important, the specific sequence does not need to match as long as all relevant items are included.

Grade student questions based on [Data], [Tasks & Student Answers].
The output json should have the format:
[{"reasoning": ...,
   "reference answer": ...,
   "comparison_with_student_answer": ...,
   "student_answer_correctness": ...},
...]

**For example:**
{Evaluation Demonstration}

---

## C  EXAMPLES OF CHARTS

We present one chart visualized in 14 different chart types used in the experiments in Figure C1 and Figure C2.

Additionally, we include our manually edited 3D chart examples in Figure C3. These charts can also be used for direct comparisons of visual element perception, such as color hue vs. color luminance in bar charts and color hue vs. texture in line charts. However, as current SOTA LMMs fail to achieve a satisfactory level of accuracy, we are unable to obtain meaningful insights at this time. We leave further exploration of LMMs' visual element understanding in 3D charts for future work.

## D  FULL RESULTS

The detailed results of four representative LMMs on 10 tasks across 14 chart types used in Section 3 and Section 4 are shown in Tables D1, D2, D3, D4.

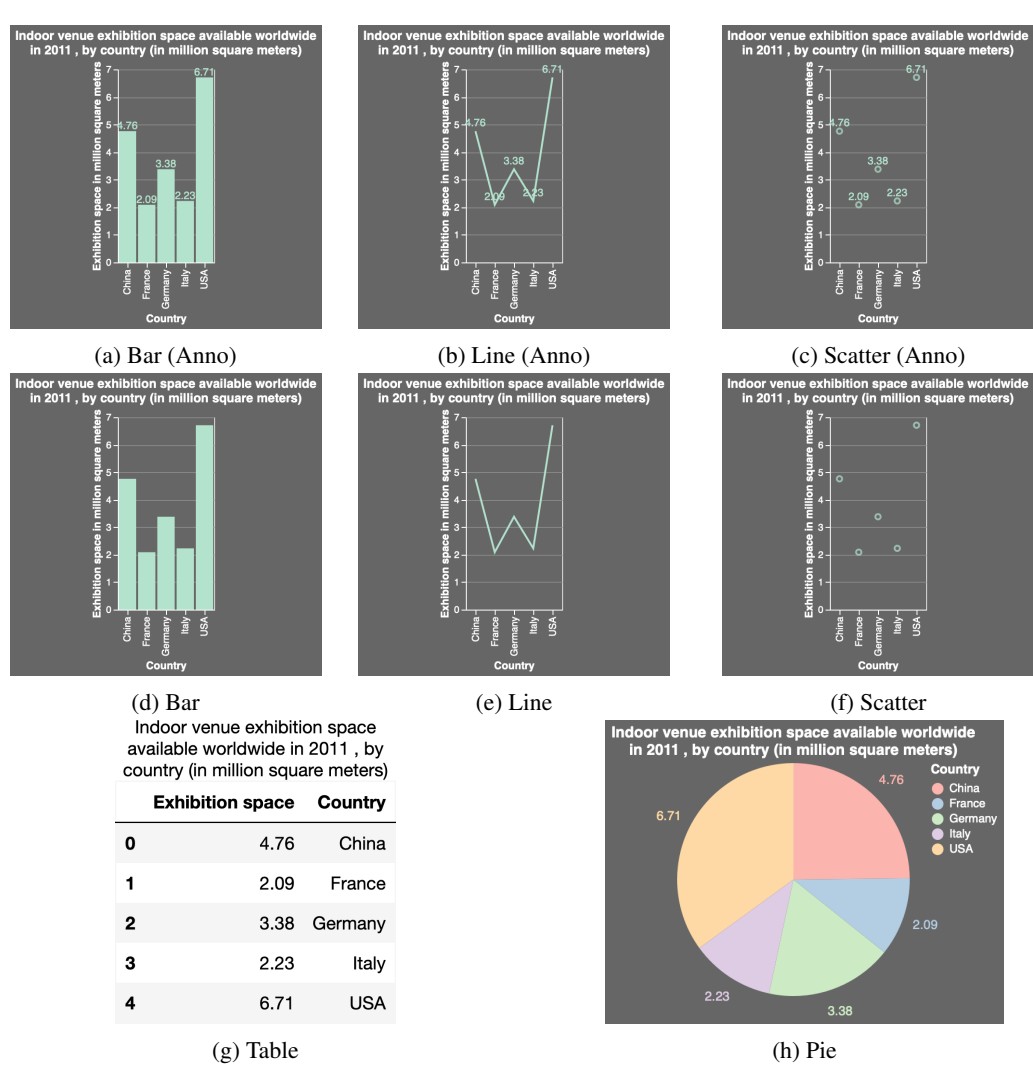

(a) Bar (Anno)  (b) Line (Anno)  (c) Scatter (Anno)

(d) Bar  (e) Line  (f) Scatter

(g) Table  (h) Pie

Figure C1: Cases of charts with and without numerical annotations. The Table is used as image input for models being evaluated.

|  | w/ Number Annotated | | | | | w/o Number Annotated | | | Single Element | | | Multiple Elements | | |
|---|---|---|---|---|---|---|---|---|---|---|---|---|---|---|
|  | Bar | Line | Scatter | Pie | Table | Bar | Line | Scatter | ↔ | ■ | ● | (↔, ★) | (●, ★) | (↔, ■, ●, ★) |
| T1 | **88.3** | 75.2 | 74.7 | 68.2 | 73.8 | 63.5 | 50.0 | 52.5 | 14.9 | 20.1 | 16.6 | 17.6 | 17.9 | 27.4 |
| T2 | **87.4** | 64.3 | 63.6 | 54.6 | 60.7 | 65.9 | 48.3 | 40.6 | 25.3 | 30.6 | 37.4 | 35.0 | 35.5 | 44.2 |
| T3 | **87.0** | 77.0 | 78.0 | 64.0 | 75.1 | 69.1 | 59.9 | 55.5 | 18.1 | 21.3 | 27.3 | 29.7 | 32.3 | 37.0 |
| T4 | 90.6 | 89.5 | 92.5 | **94.3** | 86.5 | 46.8 | 46.2 | 41.8 | 1.6 | 3.0 | 0.7 | 1.9 | 1.6 | 1.7 |
| T5 | 87.5 | 85.6 | 86.0 | 80.0 | **89.5** | 75.1 | 77.5 | 72.3 | 55.6 | 66.3 | 68.5 | 63.6 | 71.3 | 66.8 |
| T6 | 77.6 | 79.4 | **84.1** | 77.9 | 74.3 | 21.0 | 15.4 | 16.5 | 7.0 | 5.3 | 4.9 | 7.4 | 6.5 | 7.4 |
| T7 | **81.1** | 62.3 | 69.4 | 60.8 | 70.5 | 43.5 | 30.5 | 33.9 | 13.9 | 17.8 | 20.2 | 19.6 | 23.1 | 25.5 |
| T8 | **78.3** | 41.9 | 45.4 | 46.3 | 61.6 | 36.0 | 14.1 | 14.6 | 1.4 | 3.9 | 3.8 | 4.3 | 3.2 | 5.6 |
| T9 | **85.4** | 71.6 | 73.5 | 61.7 | 70.6 | 60.6 | 50.3 | 44.3 | 27.9 | 32.4 | 35.6 | 39.1 | 42.4 | 47.2 |
| T10 | **87.6** | 80.0 | 84.5 | 84.2 | 87.4 | 55.6 | 39.5 | 44.4 | 11.5 | 13.1 | 12.8 | 11.7 | 12.8 | 15.1 |
| Overall | **85.0** | 72.6 | 75.1 | 69.1 | 74.7 | 53.4 | 42.8 | 41.2 | 17.6 | 21.1 | 22.6 | 22.9 | 24.4 | 27.7 |

Table D1: All results of GPT-4o (OpenAI, 2024) on 14 types of charts across 10 task types. The best result on each task is marked in bold.

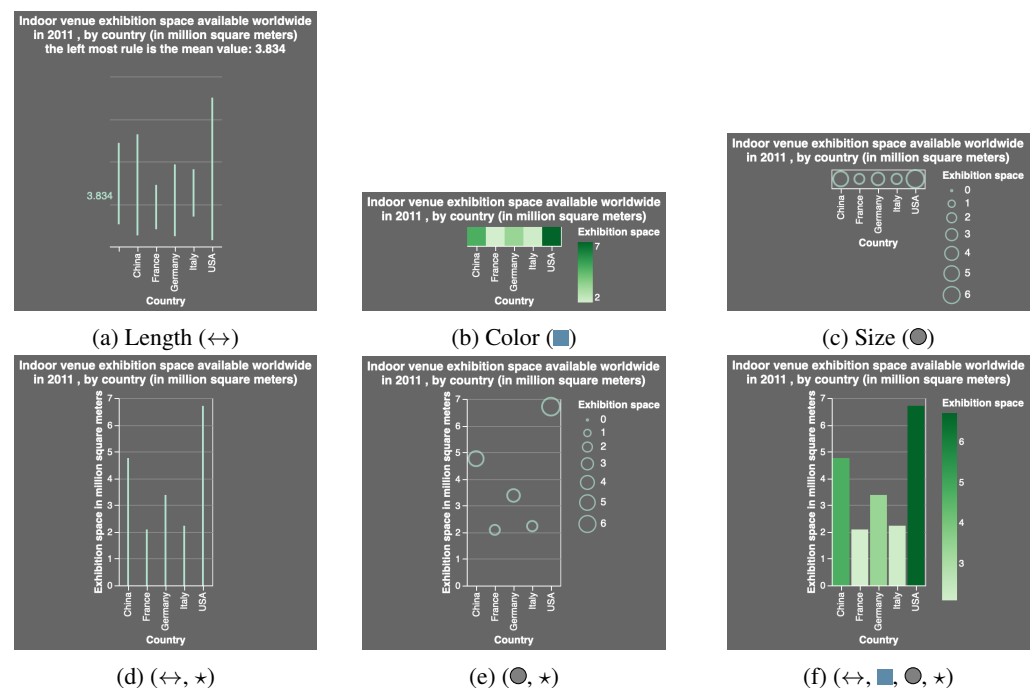

Figure C2: Cases of charts with single and multiple visual elements.

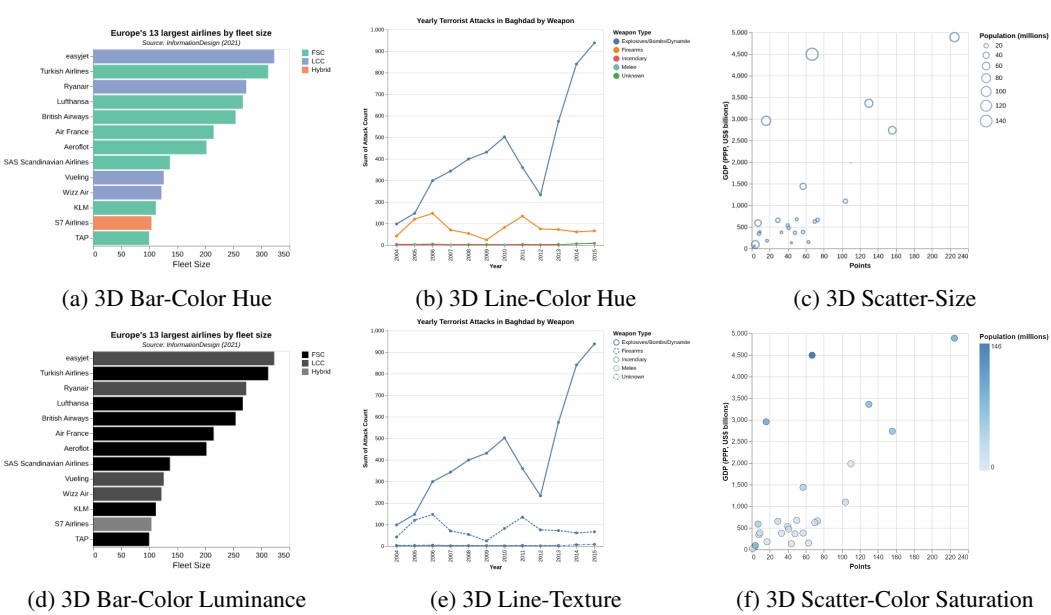

Figure C3: Cases of charts with three data dimensions (i.e., three different data attributes).

| | w/ Number Annotated | | | | | w/o Number Annotated | | | Single Element | | | Multiple Elements | | |
|---|---|---|---|---|---|---|---|---|---|---|---|---|---|---|
| | Bar | Line | Scatter | Pie | Table | Bar | Line | Scatter | ↔ | ■ | ● | (↔,★) | (●,★) | (↔,■,●,★) |
| T1 | 72.3 | 65.4 | 66.1 | 61.9 | **84.5** | 51.9 | 38.9 | 40.5 | 33.7 | 40.9 | 37.3 | 33.7 | 40.9 | 40.5 |
| T2 | 77.3 | 50.8 | 53.0 | 49.5 | **92.0** | 75.3 | 49.7 | 46.5 | 26.1 | 23.3 | 30.4 | 26.1 | 23.3 | 46.5 |
| T3 | 51.3 | 44.3 | 44.5 | 32.7 | 51.8 | **52.4** | 43.5 | 39.3 | 11.7 | 15.5 | 18.8 | 11.7 | 15.5 | 39.3 |
| T4 | 69.7 | 63.8 | 66.6 | 51.4 | **75.4** | 27.3 | 26.0 | 21.5 | 5.7 | 16.1 | 6.2 | 5.7 | 16.1 | 21.5 |
| T5 | 49.6 | 45.0 | 53.5 | 37.8 | **53.2** | 42.6 | 45.0 | 45.5 | 41.6 | 42.0 | 42.0 | 41.6 | 42.0 | 45.5 |
| T6 | 72.1 | 64.8 | 68.5 | 59.8 | **73.5** | 52.6 | 37.4 | 40.8 | 23.6 | 23.0 | 23.7 | 23.6 | 23.0 | 40.8 |
| T7 | 40.9 | 30.4 | 31.2 | 25.4 | **50.8** | 35.8 | 20.6 | 24.0 | 8.1 | 10.1 | 12.6 | 8.1 | 10.1 | 24.0 |
| T8 | 58.2 | 27.8 | 32.6 | 47.0 | **66.4** | 50.4 | 15.4 | 16.5 | 4.3 | 5.8 | 6.3 | 4.3 | 5.8 | 16.5 |
| T9 | 34.9 | 32.2 | 34.7 | 26.0 | **39.0** | 34.8 | 27.3 | 30.4 | 14.4 | 17.3 | 20.9 | 14.4 | 17.3 | 30.4 |
| T10 | 43.6 | 35.6 | 35.8 | 29.4 | **49.7** | 29.7 | 22.3 | 26.6 | 12.8 | 13.1 | 11.9 | 12.8 | 13.1 | 26.7 |
| Overall | 57.6 | 46.4 | 49.0 | 42.5 | **64.2** | 45.9 | 33.0 | 33.4 | 18.3 | 20.7 | 21.1 | 25.2 | 24.1 | 26.5 |

Table D2: All results of InternVL2 (Chen et al., 2024b) on 14 types of charts across 10 task types. The best result on each task is marked in bold.

| | w/ Number Annotated | | | | | w/o Number Annotated | | | Single Element | | | Multiple Elements | | |
|---|---|---|---|---|---|---|---|---|---|---|---|---|---|---|
| | Bar | Line | Scatter | Pie | Table | Bar | Line | Scatter | ↔ | ■ | ● | (↔,★) | (●,★) | (↔,■,●,★) |
| T1 | 70.5 | 62.7 | 63.2 | 53.6 | **85.9** | 36.3 | 26.8 | 34.7 | 32.2 | 36.7 | 32.1 | 32.0 | 37.5 | 47.0 |
| T2 | 72.7 | 53.6 | 52.9 | 30.7 | **90.3** | 59.7 | 38.5 | 40.6 | 21.0 | 22.6 | 26.1 | 31.2 | 28.5 | 33.6 |
| T3 | 39.7 | 38.2 | 36.8 | 24.3 | **47.4** | 37.6 | 38.9 | 30.6 | 10.8 | 11.0 | 13.7 | 17.8 | 12.3 | 15.0 |
| T4 | **70.6** | 67.4 | 66.5 | 51.2 | 54.8 | 14.6 | 13.1 | 14.3 | 8.9 | 15.3 | 9.0 | 11.7 | 9.6 | 10.8 |
| T5 | 62.5 | 60.0 | **64.4** | 42.0 | 62.3 | 54.5 | 54.6 | 57.3 | 53.9 | 48.7 | 50.9 | 55.3 | 50.7 | 53.5 |
| T6 | 66.8 | 62.4 | 66.1 | 53.5 | **76.4** | 27.6 | 18.7 | 27.1 | 19.2 | 17.5 | 17.8 | 17.4 | 19.8 | 21.3 |
| T7 | 37.2 | 27.5 | 28.5 | 19.3 | **50.6** | 19.5 | 15.7 | 17.5 | 5.9 | 7.4 | 9.6 | 10.3 | 10.7 | 10.7 |
| T8 | 40.7 | 24.1 | 27.8 | 21.6 | **73.7** | 27.5 | 10.3 | 15.1 | 6.6 | 7.6 | 9.3 | 7.4 | 9.9 | 8.0 |
| T9 | 24.5 | 21.3 | 22.2 | 14.3 | **27.1** | 21.5 | 25.6 | 21.1 | 10.8 | 10.7 | 14.4 | 15.0 | 15.3 | 13.3 |
| T10 | 32.1 | 26.3 | 29.8 | 14.5 | **41.5** | 21.6 | 17.2 | 19.7 | 9.6 | 12.2 | 13.8 | 13.7 | 15.1 | 14.2 |
| Overall | 52.2 | 44.7 | 46.2 | 32.7 | **61.6** | 32.4 | 26.0 | 27.9 | 17.8 | 18.9 | 19.6 | 21.2 | 20.9 | 22.7 |

Table D3: All results of Phi-3.5 (Abdin et al., 2024) on 14 types of charts across 10 task types. The best result on each task is marked in bold.

| | w/ Number Annotated | | | | | w/o Number Annotated | | | Single Element | | | Multiple Elements | | |
|---|---|---|---|---|---|---|---|---|---|---|---|---|---|---|
| | Bar | Line | Scatter | Pie | Table | Bar | Line | Scatter | ↔ | ■ | ● | (↔,★) | (●,★) | (↔,■,●,★) |
| T1 | 54.4 | 42.5 | 43.6 | 32.8 | **55.0** | 32.8 | 23.1 | 23.6 | 9.1 | 17.9 | 14.4 | 37.8 | 28.3 | 37.3 |
| T2 | **70.2** | 46.1 | 43.2 | 28.0 | 42.9 | 70.1 | 42.6 | 44.4 | 22.9 | 24.0 | 25.2 | 34.7 | 30.2 | 35.8 |
| T3 | 47.9 | 35.7 | 34.3 | 28.7 | 34.0 | **49.9** | 38.7 | 38.5 | 12.2 | 12.1 | 12.1 | 19.6 | 16.3 | 16.4 |
| T4 | **25.7** | 20.7 | 22.3 | 11.1 | 21.6 | 8.2 | 6.6 | 6.9 | 0.7 | 2.5 | 1.5 | 2.6 | 1.3 | 1.9 |
| T5 | 54.6 | 54.9 | 49.3 | 46.9 | 44.1 | 56.2 | **58.5** | 52.7 | 41.6 | 47.7 | 45.0 | 46.2 | 49.3 | 44.8 |
| T6 | 28.1 | 21.5 | 20.9 | 18.1 | 15.7 | **32.1** | 24.0 | 25.5 | 7.3 | 7.0 | 9.3 | 13.2 | 12.6 | 13.3 |
| T7 | 22.9 | 17.0 | 16.8 | 12.1 | 17.2 | **23.5** | 18.7 | 16.6 | 8.8 | 9.3 | 9.7 | 14.7 | 11.9 | 12.0 |
| T8 | 14.7 | 5.9 | 5.2 | 4.1 | **29.6** | 14.4 | 3.7 | 3.9 | 0.8 | 1.1 | 2.6 | 1.4 | 2.2 | 1.0 |
| T9 | **29.2** | 25.2 | 22.2 | 19.4 | 15.5 | 28.7 | 27.1 | 25.0 | 11.6 | 9.7 | 8.4 | 12.8 | 11.9 | 13.6 |
| T10 | 14.5 | 13.8 | 12.2 | 10.6 | 10.0 | **17.5** | 13.5 | 14.2 | 8.2 | 9.3 | 7.2 | 9.2 | 10.8 | 8.7 |
| Overall | **36.6** | 28.4 | 27.0 | 21.2 | 28.5 | 33.9 | 25.9 | 25.4 | 12.3 | 14.0 | 13.6 | 19.2 | 17.4 | 18.5 |

Table D4: All results of ChartAssistant (Meng et al., 2024) on 14 types of charts across 10 task types. The best result on each task is marked in bold.

