# OpenReview forum: "Evaluating Graphical Perception of Large Multimodal Models"
_ICLR.cc/2025/Conference — ICLR 2025 Conference Withdrawn Submission_

### Official Review · Reviewer_EpMZ · 2024-10-30

**Soundness:** 2
**Presentation:** 2
**Contribution:** 2
**Rating:** 3
**Confidence:** 4

**Summary:**

This paper provides an in-depth evaluation of the LMMs'  graphical perception ability utilizing the theory of graphical perception. They first create a dataset encompassing different types of charts and different questions with the help of GPT4o . Then, they test the SOTA LMMs with the help of an accurate GPT4o evaluation. Their results reveal several limitations, including generalizing across chart types, understanding fundamental visual elements, and reference values within a chart.

**Strengths:**

1. This paper conducts thorough analyses to assess the impact of various combinations of visual elements, chart types, and annotations on the same data.
2. This paper highlights several limitations of current LMMs and can guide future LMM designs.
3. This paper proves a data generation pipeline that can be applied to generating diverse test cases focusing on different abilities of chart understanding.

**Weaknesses:**

1. This paper tests only four LMMs. However, these models are trained on different data and resolutions, making a direct comparison impossible. They should evaluate more LMMs to enhance the reliability of their results.
1. Section 5.1 lacks many essential details, such as the threshold used to measure the LMM's ability to correctly identify important chart regions and how to annotate the ground truth region. The method may also have several flaws. For example, the importance of specific regions does not necessarily mean removing them will harm understanding. For instance, you can omit the label 2010 in Figure 4, and people will still understand that the middle bar represents the year 2010.
1. Although the paper provides some experiment results, this paper gives far more claims than the results can support. For example, in line 289, the paper claims that lightweight LMMs have weaker generalization abilities than larger models when faced with charts lacking explicit numerical cues. However, testing only one lightweight LMM means you cannot guarantee that this result can be generalized to other models.
1. Typo: line 179 LLaVa should be LLaVA.

**Questions:**

1. How can you combine position, length, and size in a bar chart at line 319? The toy chart is similar to the bar chart  that only uses position and length. Additionally, I couldn't find an example in Figure C2 that demonstrates a chart combining position, length, and size.
2. In line 414, how do you define the ground truth labeled regions and check whether the model correctly identifies a region? Do you do it by checking IoU above a threshold?
3. In Figure 5, the region containing 0.827 also shows a large correspondence with the response; how can you explain this phenomenon? Is it consistent with your analyses?

---

> ### Author Response · Authors · 2024-11-12
>
> Thanks for your valuable comments and questions.
>
> 1. Only four LMMs: In our pilot exploration, we explored llava1.5, llava1.6, Yi, etc., but we found their results were not promising compared to the selected four models. Also, to support fine-grained model analysis, we believe the current selection standard has its own merits.
>
> 2. Thanks. "Most" means 50%; we will add these details in our later revisions. You are correct that we may omit some parts of the annotated regions and still get the correct information. That’s why we only ask for most of the annotated regions.
>
> 3. Thanks; we will use softer tones in our later revision.
>
> 4. Thanks for the typo; we will fix it.
>
> Questions:
> 1. As we said in the caption of Table 3 (L312-314): "For example, the size, top part, and length of a bar are all proportional to the values."
>
> 2. Yes, intersection over union >= 50%. We will add these details in our revision.
>
> 3. Great question. First, it does not contradict what we intended to show in this figure: InternVL2 does not cover the necessary regions, so it is incorrect. Second, being sensitive to non-important regions can be the reason why the model is wrong.

---

### Official Review · Reviewer_y58c · 2024-11-04

**Soundness:** 2
**Presentation:** 2
**Contribution:** 2
**Rating:** 3
**Confidence:** 4

**Summary:**

Propose a benchmark for graphical perception abilities of large multimodal models (LMMs). The benchmark is able to eval LMMs' graphical perception in chart, visual element, and pixel levels.

**Strengths:**

The paper designed a benchmark, which has source data coupled with code to Generate the chart, enable future research to modify the code for differnt use and benchmarking

The paper systematically answered three research questions:
1. Can LMMs generalize to different chart types, the answer is No. The author also finds that numerical annotations are very important for LMMs to understand charts
2. Can LMMs learn generalizable graphical perception? The answer is No. LMMs merely follow specific and superficial perception patterns for common charts such as scatter and bar
3. Where LLMs bad at pixel-level perception? The answer is that LMMs fail to cross reference the specific values

**Weaknesses:**

1. The proposed benchmark is toy-level, not diverse enough. Though the authors aim to create a simple benchmark with better flexibility. I am afraid that only limited insights are gained with with new benchmark.
2. All three findings are not surprising, a lot of work in this field has revealed the poor perception of LMMs like MMVP, Blink. It is even harder for LMMs to reason in a Chart.
3. Given the fact that LMMs do not have a vision reasoning capability is so obvious. I think it is more improtant if the authors can come up with some methods to help LMMs with it, like the cross reference problem can be probably solved by some visual-COT methods.

**Questions:**

In summary, I appreciate the systematically analysis about the chart understanding questions. But given the fact that all findings are not surprising, and also quite explored in the wider vision community, I think the contribution is limited. What's more, apart from those analysis, author do not try anything on solving those problem, this also limits the contribution

---

> ### Author Response · Authors · 2024-11-12
>
> Thank you for reviewing our work and providing comments.
>
> 1. Diversity and limited insights. We kindly disagree with your statement. Diversity may include various dimensions, like data source, chart types, and data number.
> The data source is from various domains, as detailed in L104. We include 14 chart types and 3D charts, and our evaluation framework clearly can support more diverse chart types as well as various data numbers.
> Also, we believe more fine-grained insights can be obtained from our evaluation framework and benchmark due to the reasons provided in the introduction.
>
> 2. Thank you for the comment. We agree that works like Blink and MMVP have shown the poor perception issue, but they do not provide fine-grained insights or findings, as we explained in the introduction. Additionally, they mainly consider natural images, so it is unknown whether their findings are transferable to chart images.
>
> 3. Thank you. Due to limited space, we will leave it for future work.

---

### Official Review · Reviewer_U3Xx · 2024-11-05

**Soundness:** 3
**Presentation:** 3
**Contribution:** 2
**Rating:** 5
**Confidence:** 3

**Summary:**

This paper argues that existing benchmarks do not provide fine-grained insights into the performance of LMMs on chart-related tasks. To address this, it proposes an evaluation framework with automated task generation and response evaluation. The framework assesses LMMs at the chart, visual element, and pixel levels, revealing certain limitations of current state-of-the-art LMMs.

**Strengths:**

This paper explores the basic graph perception capabilities of current LMMs and presents several interesting observations. It emphasizes the limitations of current LMMs on simple graph perception tasks and highlights the substantial impact of annotations on model performance.

**Weaknesses:**

Dataset description: The dataset details are vague; providing more comprehensive statistics would improve clarity.

Limited contribution: Although the paper aims to design the simplest possible tasks for evaluation (line 96-100), prior work (e.g., [1]) already includes similar tasks, such as descriptive questions on information extraction and enumeration. A detailed comparison with other graphical perception benchmarks would clarify the core differences.

Lack of in-depth analysis:
1.In Section 4, the authors mention that “the addition of redundant visual elements often hurts model performance.” This is a counterintuitive yet interesting observation. More experiment analysis is needed to interpret it.

2.The paper underscores the impact of annotation on performance.However, the role of visual elements remains unclear.
It would be insightful if the authors analyzed more samples that are accurately classified without annotation. This may provide more insights into the contributions of various visual elements.

[1] Zirui Wang, Mengzhou Xia, Luxi He, Howard Chen, Yitao Liu, Richard Zhu, Kaiqu Liang, Xindi Wu, Haotian Liu, Sadhika Malladi, Alexis Chevalier, Sanjeev Arora, and Danqi Chen. Charxiv: Charting gaps in realistic chart understanding in multimodal llms. In Proceedings of NeurIPS, 2024.

**Questions:**

refer to weaknesses

---

> ### Author Response · Authors · 2024-11-12
>
> Thanks for taking time to review our work and providing comments.
>
> 1. Dataset description: We would highlight that our major contribution is the evaluation framework, and the benchmark is a byproduct. For details, please refer to Figure 2: 14 chart types, 10 questions, and 1000 datasets, resulting in 140,000 inputs for each model.
>
> 2. Limited contribution: We've explained our differences in L35-40 of our introduction.
>
> 3. Lack of in-depth analysis: Thanks for the suggestion. We explained this observation in the summary of Section 4, L345-346. We would appreciate it if you have concrete suggestions.
>
> 4. The results in Table 3 are all without annotations. The number of figures without annotations is actually larger than that of figures with annotations.
>
> Thanks again for your comments, let us know if you have any other questions.

---

### Official Review · Reviewer_5uqS · 2024-11-05

**Soundness:** 3
**Presentation:** 3
**Contribution:** 2
**Rating:** 6
**Confidence:** 3

**Summary:**

The paper explores chart understanding - one of the pain-points of modern VLMs and proposes a comprehensive evaluation framework, rooted in theory of graphical perception and targeting different levels of perception. The paper tests a mix of open and closed models and concludes that the models struggle with generalization across chart types and element patterns, and imprecisely located important regions related to a query.

**Strengths:**

- Problem - The problem is intuitively well-known, however there is a lack of benchmarks that target specific failures in understanding versus QA, thus the paper is a significant contribution to the field.
- Diversity of tasks - The tasks cover a wide range of needs and can reasonably be expected to evaluate predicted normal usage.
- Selection of Models - Good mix of open and closed models

**Weaknesses:**

- Heavy reliance on GPT for evaluation, especially in an area where LLMs are known to struggle and lack of human evaluation.
- Lack of verification of Vega-Lite outputs, since some existing datasets are known to have wrongly generated charts in them, potentially throwing all subsequent steps under question.
- Lack of case studies, makes it hard for the viewer to understand actual performance of the models.

**Questions:**

- Have you had a chance to examine the dataset in greater detail, and whether the quality of the samples?
- Any insights on how these findings would translate to real or more complex charts?
- Would you expect any differences in the results when using a more/less popular tool (compared to Vega-Lite)?

---

> ### Author Response · Authors · 2024-11-12
>
> Thanks for reviewing our work and providing valuable comments and questions.
>
> 1. GPT evaluation. In fact, GPT-4o is good at solving these tasks when the data is provided in text-only format, especially when the dataset size is small, and we manually check the accuracy of evaluation (99.5%) in Section 2.4.
>
> 2. Thanks for bringing this up. We didn't systematically test that, but during our hard-coded generation process, we barely observed any incorrect charts. We will include the details in the next version.
>
> 3. Figures 4 and 5 provide some case studies, including both responses and our explanations. We will try to include more case studies in the appendix in the future.
>
> Questions:
> 1. Please refer to our response to Weakness 2.
> 2. Thanks for the question. Given the poor performance on our simplest charts, I believe the decent performances on even more complex charts are more likely due to data contamination or relatively simple questions.
> 3. That's a great question. We don't think charts generated with different tools will result in significant performance differences based on our early exploration (We qualitatively tried charts generated with TikZ, Matplotlib, and Vega).
>
> Thanks again for these insightful suggestions and questions. We will address these concerns accordingly in our next revision.

---

### Official Review · Reviewer_nTrd · 2024-11-06

**Soundness:** 3
**Presentation:** 3
**Contribution:** 2
**Rating:** 3
**Confidence:** 4

**Summary:**

This study investigates the graphical element perception capabilities of Multi-modal Language Models (MLLMs) in chart understanding tasks. Through performing evaluations on several top-performing models on the framework, this study finds that these models struggle to generalize in chart-related tasks and even have difficulty understanding basic visual elements in chart. This study also clarifies the direction for future automation in synthesizing relevant visual chart data to improve the graphical perception and general low-level visual reasoning of MLLMs.

**Strengths:**

1. This paper presents an evaluation framework that conducts a highly detailed analysis and exploration of the reasons why current MLLMs perform poorly on chart-related tasks.  It also offers three interesting insights to guide future improvements of multi-modal large models in chart-related tasks.

2. This paper is well-organized and clearly written. The proposed pipeline data creation approach and the new dataset would benefit further research.

**Weaknesses:**

1. The paper indeed identifies the deficiencies of MLLMs in understanding table-related capabilities and provides a very detailed analysis. However, I believe that, building on these findings, there should be a deeper exploration into the essence of the holy grail problem—specifically, why MLLMs perform poorly in recognizing visual elements of chart. This could involve further analysis in terms of interpretability, training data, and model architecture. Additionally, this study should propose at least one method for automatically constructing such data to address the issues with MLLMs introduced in this paper.

2. I notice that the paper mentions that an increase in chart visual elements leads to model performance degradation. Could you further explore the impact of reasoning about relationships among multiple visual elements in charts on the performance of MLLMs, as well as their relationship to the understanding of individual visual elements?

**Questions:**

1. I noticed that the data source includes domains such as sports, news, finance, and health. Do MLLMs perform differently on similar tables across different domains? Additionally, for the same table, do different instructions significantly impact the understanding of these visual elements?

---

> ### Author Response · Authors · 2024-11-12
>
> Thanks for the comments.
>
> 1. Thanks for the suggestion. We will consider that in our future work. In terms of constructing such data, our evaluation framework can easily generate more questions and chart types, which can be used for training, as discussed in the conclusion section.
>
> 2. In the 2-dimensional charts shown in Section 4, all visual elements are used to represent values, as stated in the caption of Table 3 (L312-314): "For example, the size, top part, and length of a bar are all proportional to the values." Ideally, the models should use the most effective visual elements for specific tasks to improve accuracy; however, they are only able to perceive specific combinations of visual elements.
>
> Questions:
> 1. That's a good question. We didn't quantitatively test that, but our observation is that the differences among MLLMs in different domains are not substantial. The major issue remains generalization to different chart types and combinations of visual elements.

---

### Note · Authors · 2024-11-12

I have read and agree with the venue's withdrawal policy on behalf of myself and my co-authors.